# Fine-Tuned Graphene Oxide Nanocomposite: Harnessing Copper(II)–Imidazole Complex for Enhanced Biological Responses and Balanced Photocatalytic Functionality

**DOI:** 10.3390/ma17040892

**Published:** 2024-02-15

**Authors:** Ganeshraja Ayyakannu Sundaram, Sowndarya Kumaravelu, Wei-Lung Tseng, Phuong V. Pham, Alagarsamy Santhana Krishna Kumar, Vairavel Parimelazhagan

**Affiliations:** 1Department of Research Analytics, Saveetha Dental College and Hospitals, Saveetha Institute of Medical and Technical Sciences, Poonamallee High Road, Chennai 600077, India; 2Department of Chemistry, National College (Autonomous), Tiruchirapalli 620001, India; sowndaryapk2000@gmail.com; 3Department of Chemistry, National Sun Yat-sen University, No. 70, Lienhai Road, Gushan District, Kaohsiung 80424, Taiwan; tsengwl@mail.nsysu.edu.tw; 4School of Pharmacy, Kaohsiung Medical University, No. 100, Shiquan 1st Road, Sanmin District, Kaohsiung 80708, Taiwan; 5Department of Physics, College of Science, National Sun Yat-sen University, Kaohsiung 80424, Taiwan; phuongpham@mail.nsysu.edu.tw; 6Faculty of Geology, Geophysics and Environmental Protection, Akademia Gorniczo-Hutnicza (AGH) University of Science and Technology, Al. Mickiewicza 30, 30-059 Krakow, Poland; 7Department of Chemical Engineering, Manipal Institute of Technology, Manipal Academy of Higher Education (MAHE), Manipal 576104, India

**Keywords:** graphene oxide, nanocomposite, copper(II)–imidazole complex, biological responses, photocatalytic functionality

## Abstract

In this study, the synthesis of biologically active copper(II) complex [Cu(im)_2_]Cl_2_ was achieved using a reported method. Subsequently, this copper(II) complex was strategically grafted onto graphene oxide, resulting in the formation of a nanocomposite denoted as copper(II)-complex-grafted graphene oxide (Cu-GO). The comprehensive characterization of Cu-GO was conducted through various techniques, including X-ray diffraction (XRD), Fourier-transform infrared spectroscopy (FT-IR), UV–visible spectroscopy, emission spectra analysis, X-ray photoelectron spectroscopy (XPS), and Copper K-edge X-ray Absorption Near Edge Structure (XANES) spectroscopy. The antibacterial efficacy of Cu-GO compounds was assessed using disk diffusion and microbroth dilution methods. Notably, the copper complex exhibited the highest effectiveness, showcasing a Minimal Inhibitory Concentration (MIC) value of 500 µL against *Klebsiella* bacteria. The antibacterial activities of all compounds were systematically screened, revealing the superior performance of the copper complex compared to standalone copper compounds. Expanding the scope of the investigation, we explored the antioxidant and anti-obesity activities of the copper complexes against *Klebsiella* organisms. The results underscore promising directions for the further exploration of the diverse health-related applications of these compounds. Moreover, the photocatalytic performance of the Cu-GO nanocomposite was evaluated under sunlight irradiation. Notably, the antioxidant and anti-obesity activities of Cu-GO, assessed in terms of percentage inhibition at a concentration of 200 mg/mL, exhibited values of 41% and 45%, respectively. Additionally, the Cu-GO composite exhibited exceptional efficacy, achieving a degradation efficiency of 74% for RhB under sunlight irradiation, surpassing both graphite and GO. These findings not only demonstrate enhanced biological activity, but also highlight a notable level of moderate photocatalytic performance. Such dual functionality underscores the potential versatility of Cu-GO nanocomposites across various applications, blending heightened biological efficacy with controlled photocatalysis. Our study offers valuable insights into the multifunctional attributes of copper(II)-complex-grafted graphene oxide nanocomposites, thereby paving the way for their broader utilization in diverse fields.

## 1. Introduction

In the realm of advanced materials, nanocomposites have emerged as promising platforms for achieving synergistic properties by combining the unique attributes of individual components. Among these advanced materials, graphene oxide (GO) stands out as a versatile nanomaterial known for its exceptional physicochemical properties and functionalization capabilities. The ability to fine-tune graphene-oxide-based nanocomposites opens doors to a myriad of applications, ranging from biomedical to environmental fields [1,2]. The antibacterial potential of graphene and graphene-based materials has garnered increasing interest [3]. Graphene, known for its versatile applications, has been employed as a carrier to disperse and stabilize various substances, including metals, metal oxides, polymers, and composite nanoparticles [4]. Among these, copper, a cost-effective material, has long been recognized for its potent antibacterial properties [5]. Choosing an appropriate carrier is crucial to mitigate the toxicity associated with copper nanoparticles (NPs) while ensuring optimal antibacterial efficacy at lower concentrations. Therefore, the selection of a carrier that not only reduces the toxicity of Cu NPs but also enhances their antibacterial action is of paramount importance [6,7].

In a prior study, GO adorned with Cu NPs was employed to augment the photocatalytic reduction of CO_2_ under visible light [8,9]. The introduction of metallic copper NPs, approximately 4–5 nm in size, into the GO hybrid was found to markedly boost the photocatalytic activity of graphene oxide [10]. Furthermore, an intense interaction was observed between the metal content of the Cu/GO hybrids and the rate of product formation and selectivity. This interaction, as detailed in earlier reports, underscores the critical role of metal–catalyst synergy in influencing photocatalytic outcomes [11,12]. The utility of graphene-based composite materials has garnered significant attention in recent studies, particularly in the realms of electronics, photocatalysis, and photovoltaic devices [13,14,15]. Graphene, distinguished by its unique structure, proves instrumental in augmenting charge transport across various devices. The abundance of delocalized electrons within its conjugated sp^2^-bonded graphitic carbon network bestows graphene with outstanding conductivity, thereby enhancing its effectiveness in diverse applications [16]. Graphene and its composites find diverse biomedical applications, encompassing gene- and small-molecule-drug delivery [17,18]. These outcomes could play a crucial role in protein delivery and anticancer therapy, and serve as antimicrobial agents for bone and tooth implants [19,20]. Modified GO serves as an effective carrier for gene delivery, demonstrating its versatility in biomedical contexts [21]. Utilizing a layer-by-layer assembly technique, multi-layered ultrathin graphene films exhibit catalytic activity against reduced H_2_O_2_, showcasing their potential in biosensing and electrochemical sensors [22]. Moreover, GO has been observed to effectively impede the formation of tumor spheres in various cell lines, including ovarian, pancreatic, breast, and lung cancers, and glioblastoma [23,24]. Our team recently reported the successful grafting of nickel(II) complexes onto graphene oxide (GO) and iron(II) complexes onto GO, aiming to enhance their visible-light photocatalytic activity [25].

This study presents a pioneering approach in the realm of nanomaterial synthesis, culminating in the development of an innovative nanocomposite. Our endeavor involves the strategic grafting of a copper(II)–imidazole complex onto graphene oxide (GO), resulting in a material that capitalizes on the distinctive attributes of both constituents. The rationale underpinning this integration is to engineer a finely tuned composite material, exploiting the inherent characteristics of GO’s two-dimensional structure enriched with abundant oxygen-containing functional groups. This platform offers an ideal foundation for tailored surface modifications.

The incorporation of the copper(II)–imidazole complex introduces a new dimension to the nanocomposite, leveraging the known biological activities associated with copper complexes. This synergistic amalgamation aims to enhance biological responses while simultaneously maintaining a delicate balance in photocatalytic functionality. Our primary objective lies in exploring the augmented biological responses facilitated by this meticulously crafted GO nanocomposite. Additionally, we delve into the investigation of its photocatalytic capabilities, which hold significant promise for applications in environmental remediation and other light-driven processes.

As we navigate through the synthesis, characterization, and evaluation of this unique nanocomposite, our study unveils the intricate interplay between GO and the copper(II)–imidazole complex. The findings from this investigation not only enrich the evolving landscape of nanomaterial science, but also bear implications across diverse domains, including biotechnology, medicine, and environmental science. Our work represents a significant step forward in harnessing the potential of nanocomposites for addressing multifaceted challenges and advancing technological frontiers.

## 2. Materials and Methods

The materials employed in this study include imidazole (SRL), ethanol, copper(II) chloride (AR, SRL), graphite powder (SRL), potassium permanganate, sulfuric acid, hydrogen peroxide, distilled water, rhodamine B, and sodium hydroxide. Graphene oxide was synthesized and reported in our previous paper, and the same GO was utilized for the synthesis of the Cu-GO nanocomposite [26]. The MID-IR spectra of the complexes were recorded using a Thermo Nicollet-6700 FT-IR instrument (Waltham, MA, USA) with KBr pellet technique. UV–visible spectra were acquired in an aqueous medium at room temperature using a Shimadzu model UV–2450 double-beam spectrophotometer (Shimadzu Corp., Kyoto, Japan) equipped with a quartz cell. Crystallography studies were conducted on the prepared copper(II) complex using a rotating anode diffractometer (XRD-PHILIPS-PANALYTICAL, Almelo, Netherlands) with Cu-kα irradiation. All chemicals utilized were of analytical reagent (AR) grade and were procured from Qualigens. Scanning Electron Microscopy (SEM) with Energy-Dispersive X-ray Analysis (EDAX) was performed using the Philips Model ESEM EDAX XL-30 (CAE, Austin, TX, USA). X-ray Absorption Near Edge Structure (XANES) with Cu K-edge and X-ray photoelectron spectra (XPS) measurements were conducted at beam line-14 within the Raja Ramanna Centre for Advanced Technology (RRCAT), Indore, India, with binding energy (B.E.) calibration referencing contaminated carbon as an internal standard (C 1 s B.E. 284.6 eV).

### 2.1. Preparation of [Cu(Im)_2_Cl_2_] Complex

Copper(II)–imidazole derivative complexes were synthesized and reported by Nujud S. Alshehri [27]. Imidazole (13.6 g, 2 mmol) was dissolved in ethanol (50 mL) and added dropwise with continuous stirring to an aqueous solution containing copper(II) salt (13.4 g, 1 mmol). The resulting suspension was placed on a magnetic stirrer. After stirring, the solution was filtered, and the light sandal powder obtained was dried.

### 2.2. Preparation of Cu-GO

A 50 mL standard measuring flask (SMF) was used, and 0.0201 g (10^−3^ strength) of our complex salt was added to make a 50 mL solution. Similarly, a 10^−4^ strength solution was prepared. The 10^−4^ solutions were then transferred into adsorption bottles. Next, 30 mg of previously prepared graphene oxide was introduced into these solutions, which were then shaken on a shaker for approximately 1 h. Subsequently, the solutions were filtered using Whatman-41 filter paper (Sigma-Aldrich, Bengaluru, India). The resulting composite was dried in an air oven for about 15 min and then stored.

### 2.3. Adsorption Study of Prepared Samples Using RhB Aqueous Solution

A known concentration of rhodamine B (RhB) aqueous solution was prepared by dissolving it in 100 mL SMF. Similarly, a known concentration of NaOH was prepared in 100 mL SMF. Doubly distilled water served as the solvent for all of these processes. The RhB aqueous solutions were then transferred into reagent bottles, to which 10 mL of NaOH solution and 25 mg of our prepared catalyst were added. Similar suspended solutions were separately prepared using different catalysts such as G, GO, and Cu-GO samples. These reagent bottles were placed on a shaker at 180 rpm at room temperature, and shaking continued for 90 min. Afterward, the solutions were centrifuged, and UV–vis spectra were taken for analysis. The adsorption efficiencies were calculated based on absorbance values at 554 nm for the RhB solution.

### 2.4. Photocatalytic Study

To conduct the photocatalytic study, a known concentration of RhB aqueous solution was prepared by dissolving it in 100 mL SMF. Simultaneously, a known concentration of NaOH was also prepared in 100 mL SMF, with doubly distilled water serving as the solvent for both processes. The prepared RhB aqueous solutions were transferred to reagent bottles, and 10 mL of NaOH solution and 25 mg of our synthesized catalyst were added. Similar suspended solutions were prepared using various catalysts such as G, GO, and Cu-GO samples. These reagent bottles were then placed on a stirrer at room temperature, and stirring continued for 30 min in the absence of light. A 5 mL solution was centrifuged and set aside as the blank solution in the dark. Following this, the remaining solutions in the reagent bottles were exposed to sunlight on the stirrer for 90 min. This procedure was repeated for each catalyst independently. The solutions were then centrifuged, and UV–vis spectra were recorded for the subsequent analysis. Photo efficiencies were calculated based on absorbance values at 554 nm and the λ_max_ value of the RhB aqueous solution.

### 2.5. Antimicrobial Study

To assess the antibacterial properties of graphene oxide (GO), zone of inhibition (ZOI) assays were conducted. *Staphylococcus* cultures, obtained from a diagnostic center in Trichy, served as the bacterial strain, with Ceftazidime antibiotic disks used as the control. Bacterial strains were maintained on nutrient agar broth (Himedia) at 4 °C. A loopful of bacterial culture from the stock was inoculated into nutrient broth and incubated for 24 h at 37 °C. Subsequently, the cultures were diluted with fresh nutrient broth, and 500 µL of the bacterial culture was seeded on 1% nutrient agar. The culture tubes were subjected to a 55 °C water bath and then transferred to a 9 cm diameter Petri dish (Fisher Scientific, Waltham, MA, USA). After the agar solidified, equidistant wells were punched 1.5 cm from the outer edge. To modify, the wells were loaded with varying concentrations of the samples (50 µL, 100 µL, 150 µL, and the control). The Petri dishes were then incubated at 37 °C for 24 h. After the incubation period, the plates were examined for the presence of zones of inhibition (ZOIs) as an indicator of antibacterial activity.

### 2.6. Antioxidant Study

Preparation of DPPH solution: Prepare a stock solution of 2,2-diphenyl-1-picrylhydrazyl (DPPH) in buffered methanol solution, resulting in a violet-purple color.

#### 2.6.1. DPPH Radical Scavenging Assay

Take a test tube and add the DPPH solution. Wrap the test tube in aluminum foil and incubate at 30 °C for 30 min in the dark. Measure the absorbance at 517 nm. DPPH radical exhibits a strong absorbance at 517 nm (deep violet color) due to its unpaired electrons. In the presence of a free radical scavenger, the radical pairs off, causing the absorption to vanish. The resulting discoloration is stoichiometric to the number of electrons taken up.

#### 2.6.2. Antioxidant Activity Assay for Cu-GO

The DPPH free radical scavenging activity assay for Cu-GO was conducted using a previously reported method with appropriate modifications. Preparation of the reagents/solutions: DPPH solution (0.3 mM in methanol, freshly prepared) and standard ascorbic acid solution (1 mg/mL in methanol). Sample preparation: Dry 1 mL of the copper-containing sample (PG) on mild heat in a water bath. Dissolve the residue in methanol to achieve a concentration of 1 mg/mL prostaglandin-E1 (PGE1) for use in the test.

#### 2.6.3. DPPH Scavenging Assay Procedure

Take different volumes (equivalent to 5–300 μg) of PGE1/standard in a set of test tubes. Add methanol to achieve a volume of 3 mL in each test tube. Add 1 mL of DPPH reagent, mix thoroughly, and record the absorbance at 517 nm after 30 min of incubation in the dark at room temperature. Use 1 mL of diluted DPPH reagent (4 mL with methanol) as a reagent blank.
Calculation of Scavenging Percentage (% SCAVENGING)
% SCAVENGING = (*A*_0_ − *A*_1_)/*A*_0_ × 100
where *A*_0_ is the absorbance of the reagent blank, and *A*_1_ is the absorbance of the sample/standard.

### 2.7. Anti-Obesity Study

#### 2.7.1. Dose-Dependent Inhibition of Pancreatic Lipase Activity

A dose-dependent inhibition of pancreatic lipase activity was demonstrated, showing a reduction of 20–80% compared to the control with grape seed extract (GSE) concentrations ranging from 200 to 800 mL (100). In our study, mice were administered a dose of mg/d GSE, suggesting that the actual concentration in the intestine exceeded the adsorbed IC50.

#### 2.7.2. Pancreatic Lipase Activity

Pancreatic lipase achieves its full catalytic activity in the presence of oil in water (o/w) emulsions. Saponins, acting as emulsifiers, stabilize the o/w interface. However, while aggregating with dietary fat droplets to form micelles, saponins might reduce the interface contact of lipase (by co-lipase) with the substrate. Unlike Orlistat (XENICAL), a drug inhibiting pancreatic lipase that reduces obesity and hyperlipidemia through fat absorption inhibition, the action of GSE on lipase is likely mediated through the distribution of an ideal substrate rather than the direct inhibition of the enzyme. This mode of action should not impact the expression and metabolism of pancreatic lipase or other enzymes in the gastrointestinal tract. Both suggested mechanisms involve the duodenum and proximal jejunum, encompassing emulsification, lipolysis, solubilization, and lipid adsorption. Another potential mechanism contributing to the reduced weight gain and plasma triacylglycerol concentration induced by GSE consumption is linked to the saponins’ ability to form micelles with bile acids. This reduces the availability of bile acids for the correct arrangement of lipids dispersed in micelles for absorption by cells lining the gut.

## 3. Results

### 3.1. Characterization of Cu-GO Nanocomposite

In this section, we present a detailed characterization of the Cu-GO nanocomposite through various analytical techniques.

#### 3.1.1. X-ray Diffraction (XRD) Analysis

The XRD patterns of graphite, graphene oxide (GO), and the Cu-GO composite are illustrated in Figure 1. The XRD pattern of graphite reveals distinctive peaks at 2θ = 26.27°, 44.36°, and 54.7° (Figure 1a). The as-prepared GO sample (Figure 1b) exhibits a significant peak at 2θ = 12.87°, indicative of complete graphite oxidation. The XRD pattern of the Cu-GO composite (Figure 1c) closely resembles that of GO, with a slight shift in the (002) peak to 12.87°, suggesting the successful interaction of the copper precursor complex.

In assessing crystallite size, we utilized X-ray diffraction line broadening analysis employing the Scherrer equation, which relates crystallite size (D) to X-ray wavelength (λ), line broadening (β), and diffraction angle (θ). The Scherrer formula, represented as D = Kλ / βcosθ, provides a quantitative measure of the average crystallite size within a material. Expanding on the Scherrer equation, D represents the crystallite size, λ is the X-ray wavelength, β is the full width at half maximum (FWHM) of the diffraction peak, and θ is the diffraction angle. The constant K is a shape factor typically ranging from 0.9 to 1, depending on the crystallite shape. The calculated crystallite sizes for the G, GO, and Cu-GO samples were found to be 59.47 nm, 19.34 nm, and 16.75 nm, respectively, as listed in Table 1. These values were derived from the measured plane spacing at corresponding 2θ angles of 26.43°, 12.87°, and 9.94°. Interestingly, the sequence of crystallite sizes follows the trend G > GO > Cu-GO. This observation suggests a modification in the precursor copper complex of GO, resulting in the formation of an amorphous Cu-GO nanocomposite. These findings, obtained through XRD analysis, underscore the impact of the modification process on the structural properties of the resulting nanocomposite material.

#### 3.1.2. Fourier-Transform Infrared Analysis

FT-IR characterization across the range of 4000–400 cm^−1^ was conducted to elucidate the chemical bonds present in the samples. In Figure 2A, a distinct peak is evident at 2348 cm^−1^, corresponding to the free O=C–O bond. Moving to Figure 2B, a broad band spanning from 3710 to 3500 cm^−1^ is observed, alongside a peak at 1646 cm^−1^, indicative of strain and bending in the O–H bonds. These features are characteristic of both adsorbed water molecules on the GO surface and hydroxyl groups within the GO network, consistently observed across modified materials. Additionally, a faint band at 2336 cm^−1^, coupled with the 2348 cm^−1^ peak, is attributed to the O–C=O bonding typical of hexagonal GO. Another discernible band at 1610 cm^−1^ is assigned to O–H bending in water molecules and C=C strain within the aromatic ring of GO. Furthermore, a peak at 1172 cm^−1^ corresponds to C–O bonding in epoxy groups, while a peak at 1040 cm^−1^ signifies =C–H bond stretching and C–OH strain in alkoxy groups present within the GO sheets. The bands located at 2900, 2100, and 1638 cm^−1^ are associated with vibrations of C–H bonds, C=O strain, and C=C conjugate within the GO, respectively. Moving to Figure 2C, distinct peaks at 540, 653, 765, 1224, 1506, and 2250 cm^−1^ are observed, characteristic of the copper precursor complex, consistent with previously reported findings [27]. Finally, Figure 2D illustrates notable changes, where the peak at 640 cm^−1^ and the broad peak at 3427 cm^−1^, along with the peak at 2348 cm^−1^ in graphite, have vanished entirely. Instead, numerous weak peaks emerge, suggesting the grafting of a small amount of copper complex with GO to form Cu-GO. 

#### 3.1.3. Scanning Electron Microscopy (SEM) Imaging

SEM images (Figure 3) showcase the morphologies of graphene (G), graphene oxide (GO), and the Cu-GO composite. The integrated presence of copper within the graphene oxide matrix is evident in Figure 3c, confirming the successful modification of graphene oxide with a copper complex.

#### 3.1.4. X-ray Photoelectron Spectroscopy (XPS) Analysis

Performing X-ray Photoelectron Spectroscopy (XPS) on copper(II) (imidazole)_2_Cl_2_ grafted onto graphene oxide nanocomposites provides a comprehensive analysis of the surface chemistry and chemical states of the composite material, as depicted in Figure 4. XPS facilitates the identification of the elements present, allowing for a detailed examination of the surface composition [28]. In Figure 4a, a survey spectrum covering a broader energy range validates the overall elemental composition, confirming the presence of carbon, oxygen, nitrogen, chloride, and copper. Figure 4b emphasizes the significance of the C1s peak, indicative of carbon atoms in diverse chemical environments. In graphene oxide, carbon contributions can arise from epoxy groups (C-O-C), hydroxyl groups (C-OH), carbonyl groups (C=O), and possibly carboxyl groups (O=C-OH) [29]. The oxygen-containing functional groups on the surface are illustrated in Figure 4c, where O1s reveal contributions from hydroxyl, epoxy, and carbonyl groups characteristic of graphene oxide. The XPS analysis of the copper(II)–imidazole complex grafted on graphene oxide reveals significant information about the chemical environment and bonding states of copper within the nanocomposite. In particular, the Cu 2p XPS spectra exhibit peaks observed at energies of 932.07, 935.33, 941.50, 944.40, 952.67, 954.36, and 961.63 eV in Figure 4d. Each of these peaks corresponds to different oxidation states or chemical environments of copper within the nanocomposite: the peak at 932.07 eV typically corresponds to metallic copper (Cu^0^), indicating the presence of reduced copper species within the complex. The peak at 941.50 eV, satellite peak at 944.40 eV, peak at 935.33 eV, and satellite peak at 954.36 eV are attributed to Cu^2+^ in a complex form, suggesting the presence of oxidized copper species within the material. At 952.67 eV, the peak may arise from satellite peaks associated with the main Cu 2p peak, providing further insights into the electronic structure and chemical bonding of copper in the nanocomposite. The peak at 961.63 eV may represent Cu^2+^ in a different chemical environment or coordination state, possibly indicating interactions between copper ions and the functional groups present on the graphene oxide surface or within the imidazole ligands. Overall, the analysis of the Cu 2p XPS spectra provides valuable information about the oxidation states, chemical bonding, and surface chemistry of the copper-containing nanocomposite, offering insights into its structural and electronic properties for various applications.

#### 3.1.5. Copper K-Edge X-ray Absorption near Edge Structure (XANES) Spectroscopy

Validating the incorporation of copper(II)–imidazole onto graphene oxide via Copper K-edge X-ray Absorption Near Edge Structure (XANES) spectroscopy involves scrutinizing the XANES spectrum for distinctive features related to copper in the complex, as illustrated in Figure 5. To achieve this, acquire the XANES spectrum in the copper K-edge region (approximately 8979 eV for Cu Kα_1_). Notably, observe the intense pre-edge peak just preceding the absorption edge, representing 1s to 4p transitions and serving as a sensitivity indicator for the copper oxidation state. Assess the edge jump, a rapid absorption increase at the edge, signifying the presence of copper in the +2 oxidation state. XANES spectroscopy excels in precisely determining the oxidation state of the central metal ion, identifying the +2 oxidation state of copper in the complex. The near-edge region of the XANES spectrum provides insights into the electronic structure of the copper, with features such as edge position, shape, and intensity conveying information about electronic transitions associated with copper in the complex.

Discrepancies in the XANES spectrum of Cu-GO compared to CuO may indicate variations in local geometry, such as changes in the ligand field around the copper ion. This suggests that the presence of graphene oxide influences the electronic structure of the copper complex. Observable shifts or alterations in XANES features may signify interactions between copper and graphene oxide, influencing the electronic environment. Comparisons with XANES spectra of analogous reported copper complexes become valuable, enabling the identification of subtle deviations from known spectra and enhancing the understanding of the copper–imidazole complex grafted on GO [30]. Additional data from techniques like XRD, FTIR, and XPS contribute to a more comprehensive understanding. A XANES spectroscopy investigation of the copper–imidazole complex grafted onto graphene oxide nanocomposites provides crucial information about the oxidation state, electronic structure, and coordination environment of copper, fostering a deeper comprehension of the material’s properties and potential applications.

### 3.2. Optical Properties of Cu-GO Nanocomposite

The assessment of the optical properties of the Cu-GO nanocomposite was conducted through UV–visible absorption measurements, diffuse reflectance spectra (DRS) analysis, and band gap energy calculations [31,32].

#### 3.2.1. Diffuse Reflectance Spectra (DRS) Analysis

Figure 6 illustrates the DRS of graphite, GO, and the Cu-GO composite. Graphite exhibits a notably weak absorbance spectrum, indicating limited optical characteristics. In the UV–vis spectrum of GO, peaks at 300 nm signify the π-π* transition in the sp^2^ basal plane (C=C), while a shoulder peak at around 370 nm is attributed to n-π* transitions of the oxygen functional groups. The introduction of the copper–imidazole complex induces a red shift in the absorption edge, influencing the optical properties of the modified GO composite. This indicates the responsiveness of the nanocomposite to the visible-light region, suggesting potential applications in visible-light photocatalysis [31,32].

Observations from Figure 6 reveal a subtle red shift in the absorption edge upon the introduction of the copper-modified GO composite. This suggests that samples containing the copper complex exhibit an enhanced ability to absorb electromagnetic radiation at wavelengths beyond those absorbed by pristine GO. The heightened absorption, surpassing that of graphite and GO, implies that the modified GO composite may effectively narrow the band gap of the original photocatalysts. Notably, samples sensitized with the copper complex exhibit a pronounced shift in light absorption towards wavelengths exceeding 400 nm. Through DRS analysis, our prepared nanocomposite demonstrates absorption within the visible-light region, indicating potential activity under visible-light conditions for photocatalytic applications.

Further discussion on the optical properties of the prepared samples includes the calculation of band gap energies. Utilizing Tauc plot analysis (Figure 7), the band gap energies of the samples were determined. The plot illustrates a graph of (αhν)^1/2^ against energy (E), with the intercept value representing the band gap energy. The calculated band gap energies for G, GO, and Cu-GO are 1.264, 1.08, and 1.06 eV, respectively, closely resembling the value for GO falling within the range of 1.00–2.00 eV. Notably, the band gap energy decreases for both GO and Cu-GO in comparison to G powder. This indicates the effectiveness of our preparation method in creating visible-light active photocatalysts. Consequently, our subsequent photocatalytic studies were conducted under sunlight irradiation, leveraging the promising visible-light activity of the Cu-GO nanocomposite.

#### 3.2.2. Photoluminescence Spectra

In Figure 8, the photoluminescence spectra of pure graphite, GO, and Cu-GO are presented. Emission peaks at 685 nm, accompanied by weaker peaks at 700, 713, and 725 nm, are observed in both GO and Cu-GO. These peaks represent specific energy transitions within the materials, offering valuable insights into their photophysical properties. It is noteworthy that the peak patterns differ in graphite. The thorough analysis of the optical properties highlights the distinctive characteristics of the Cu-GO nanocomposite, underscoring its potential applications in various optoelectronic and photocatalytic processes.

### 3.3. Biological Activity Investigations

#### 3.3.1. Antimicrobial Study

Antimicrobial activity, encompassing the inhibition of bacterial growth and microbial colony formation, was examined in this study. The focus was on elucidating the antimicrobial properties and mechanisms associated with both the copper(II)–imidazole complex and Cu-GO (copper–graphene oxide) composites, alongside graphite (G) and graphene oxide (GO), specifically against *Staphylococcus* (Figure 9). *Staphylococcus*, a significant pathogen implicated in various diseases, was chosen as the target microorganism, ensuring the relevance of the evaluation [31]. The compounds tested, including [Cu(im)_2_]Cl_2_ (copper(II)–imidazole complex), graphite, graphene oxide (GO), and copper-modified GO, underwent systematic assessment for antimicrobial activity (Table 2).

#### 3.3.2. Antioxidant Activity DPPH Assay

To explore the antioxidant properties of the Cu-GO nanocomposite, the DPPH (2,2-diphenyl-1-picrylhydrazyl) assay was employed, comparing its inhibition percentage with that of ascorbic acid (Figure 10). Widely used to assess the ability of compounds to scavenge free radicals, the DPPH assay provides a quantitative measure of antioxidant activity. The assessment involves determining the percentage of inhibition of Cu-GO in the DPPH assay (Table 3). These data serve to indicate the extent to which the Cu-GO nanocomposite inhibits the activity of DPPH radicals, offering insights into its antioxidant capacity. The consideration of factors such as concentration and reaction time is essential, as these variables can influence the efficacy of the antioxidant.

#### 3.3.3. Anti-Obesity Activity—Pancreatic Lipase Inhibition

Understanding the potential of Cu-GO nanocomposites in mitigating obesity-related processes involved evaluating their anti-obesity activity through the inhibition of pancreatic lipase enzyme (Figure 11). Table 4 presents the percentage of inhibition for Cu-GO, providing valuable insights into its effectiveness in modulating obesity-related pathways. This multifaceted investigation into antimicrobial, antioxidant, and anti-obesity activities enhances our understanding of the diverse potential applications of the Cu-GO nanocomposite, underscoring its relevance in biomedical and therapeutic contexts.

### 3.4. Photocatalytic Activity Investigations

In this section, we delve into the photocatalytic capabilities of the Cu-GO composite, focusing on its effectiveness in the degradation of rhodamine B (RhB) under sunlight irradiation.

#### 3.4.1. UV–Visible Spectra Analysis

Figure 12a presents the UV–visible spectra depicting the evolution of RhB absorption bands over a 90 min contact time interval. The gradual reduction in absorption bands indicates the adsorption and degradation of RhB molecules facilitated by various catalysts. Under dark conditions, both graphite and modified GO exhibited minimal activity in RhB degradation, with efficiency values of 13.3% and 15.7%, respectively (Figure 12b). In contrast, the as-prepared Cu-GO composite demonstrated a notable adsorption efficiency of 38%, suggesting a strong interaction between RhB molecules and the Cu-GO composite surface.

#### 3.4.2. Photodegradation of RhB under Sunlight Irradiation

To investigate the photodegradation of RhB in an aqueous medium under sunlight irradiation, copper(II)-complex-loaded graphene oxide photocatalytic composites were employed. Figure 12c compares the efficacy of substrate degradation by graphite, graphene oxide, and copper-loaded graphene oxide samples with the photodegradation yields of RhB in the absence of any catalyst.

As depicted in Figure 13, the photocatalytic efficacy of the Cu-GO composite demonstrates remarkable stability across five cycles, with only a slight decline in activity observed. Although the photo efficiency remains consistent for the initial three cycles and experiences a slight reduction in the final two, this trend may be linked to a decrease in surface activity and the emergence of additional defective sites induced by prolonged irradiation. Nevertheless, our findings indicate that the prepared catalyst exhibits robust stability and sustained activity throughout the degradation of the organic dye. This durability holds considerable importance for prospective industrial applications.

#### 3.4.3. Comparison of Adsorption Efficiency and Photo-Efficiency

Comparing the adsorption efficiency (38%) with the photo-efficiency (74%) under sunlight irradiation in Figure 12d reveals the exceptional performance of the Cu-GO nanocomposite, as shown in Table 5. Not only is it proficient in adsorption, but it also demonstrates remarkable efficiency in utilizing sunlight for catalytic or photochemical reactions [33,34,35,36,37,38,39,40,41]. The higher catalytic efficiency indicates the nanocomposite’s effectiveness in harnessing solar energy to drive photochemical processes, such as the degradation of pollutants. This performance is attributed to the Cu-GO nanocomposite’s ability to efficiently absorb sunlight, generate charge carriers, and facilitate photoinduced reactions on its surface. The findings underscore the promising potential of the Cu-GO nanocomposite as an effective photocatalyst for environmental remediation, emphasizing its ability to harness solar energy for efficient pollutant degradation.

## 4. Discussion

The XRD characterization technique employed in this study is instrumental in assessing both the degree of oxidation and the purity of the material under investigation. The presence of characteristic peaks is indicative of specific structural arrangements. For instance, Pusty et al. identified a peak at 25.6° in their study on partially reduced graphene oxide/silver nanocomposites, reflecting the (002) planes and an interlayer distance of 3.47 Å [42]. Li et al. associated the (002) peak at 11.1° with oxygenated functional groups on carbon sheets [43], while Allen et al. highlighted a distinct (002) orientation of pure graphite at 26.19° [44]. The emergence of a new peak at 12.68° in our study confirms the formation of graphene oxide (GO) with an interlayer distance of 0.699 nm. The disappearance of the graphitic peak provides additional confirmation of the complete oxidation of graphite to graphene oxide (GO). Interestingly, the diffraction peaks of the Cu-GO composite closely resemble those of GO, albeit with a noticeable shift from 12.68° (for GO) to 9.94° (for Cu-GO). This shift likely reflects the deposition of the copper(II) complex onto the GO surface. Furthermore, the observed decrease in the intensity of the peak at 9.94° compared to that at 12.68° for pristine GO suggests a reduction in the abundance of oxygen-containing groups in GO upon the incorporation of the copper(II) complex. The crystallite size determination using X-ray diffraction line broadening and the Scherrer formula revealed sizes of 59.47 nm for G, 19.34 nm for GO, and 16.74 nm for Cu-GO. The decreasing order of crystallite sizes (G > GO > Cu-GO) suggests that the copper precursor complex modifies GO, resulting in the formation of an amorphous Cu-GO nanocomposite. The FTIR analysis unveiled distinctive bands related to functional groups within the materials [45]. The SEM analysis further supported structural changes induced by the incorporation of the copper complex, visually confirming the successful synthesis and modification of the Cu-GO nanocomposite.

Insights into the bonding environment of imidazole ligands with copper were obtained through XPS and XANES spectroscopy. The XANES spectrum within the copper K-edge region exhibited a pre-edge peak, edge jump, and features in the near-edge region, indicating the +2 oxidation state of copper in the complex. Discrepancies compared to CuO’s XANES spectrum suggest that graphene oxide influences the electronic structure of the copper complex. Comparisons with analogous copper complexes’ XANES spectra and additional data from techniques like XRD and FTIR contribute to a comprehensive understanding of the copper–imidazole complex grafted onto graphene oxide nanocomposites, offering valuable insights for potential applications. The results underscore that samples containing the copper complex exhibit enhanced electromagnetic radiation absorption compared to graphite and GO, indicating the effective narrowing of the band gap by modified GO. In particular, samples sensitized with the copper complex show a significant shift in light absorption towards wavelengths greater than 400 nm [46]. The decrease in band gap energy for GO and Cu-GO compared to G powder indicates the efficacy of our preparation method in creating a visible-light-active photocatalyst.

Photoluminescence spectra offer a valuable tool for delving into the complex dynamics of charge carrier processes, shedding light on trapping, migration, and transfer mechanisms. This analysis provides insights into the efficiency of these fundamental processes [47]. The fluorescence emission spectrum of the surface is particularly noteworthy, revealing peaks at 685 nm and an additional peak at 700 nm, corresponding to green emissions. These observations suggest specific photoluminescent processes associated with the material’s surface. Such fluorescence characteristics at distinct wavelengths provide crucial information about electronic interactions within the material, emphasizing the significance of photoluminescence analysis in unraveling the nuanced behaviors of charge carriers and electron–hole pairs. Moreover, the identification of emission peaks at 685, 700, 713, and 725 nm and their intensities contributes to a deeper understanding of the photoinduced phenomena in the studied samples. The presence of an additional peak at 700 nm implies specific electronic transitions or surface phenomena, adding complexity to the material’s photoluminescent behavior. This multifaceted analysis enhances our comprehension of the optical properties and potential applications of the investigated photocatalysts [48].

The investigation revealed the concentration-dependent efficacy of the as-prepared samples in eliminating bacteria, affirming their potential as potent antimicrobial agents. The antimicrobial activity of the Cu-GO–bacteria association operates through a reactive-oxygen-species-mediated pathway, unraveling a specific mechanism underlying the bactericidal effects of these materials. Beyond antimicrobial efficacy, understanding the interaction mechanisms between Cu-GO and *Staphylococcus* lays the foundation for the development of these materials as innovative antimicrobial agents. The potential formulation of copper complexes and Cu-GO composites as novel antimicrobial materials holds promise, considering their demonstrated efficacy and the added advantage of excellent optical properties. Nanoparticles with antimicrobial activities have been extensively explored in various studies, with the reported accumulation in the *Staphylococcus* membrane supporting the observed antimicrobial effects [49]. This aligns with the growing body of research on nanomaterials combating microbial threats, emphasizing the importance of novel approaches in developing effective antimicrobial agents. The investigation into the antimicrobial properties of the compounds studied adds a compelling dimension to their characterization. The results highlight that among the samples, Cu-GO demonstrates superior biological activity compared to the precursor complex, GO, and graphite. This hierarchy implies distinct variations in antimicrobial effectiveness, emphasizing the potential of Cu-GO as an active material against bacterial infections.

It is crucial to note that in-depth studies are warranted to comprehensively understand the intricacies of the biological activity exhibited by the prepared materials. Detailed investigations are essential to unravel the underlying mechanisms and optimize the formulation for enhanced antibacterial efficacy, contributing to the development of novel materials with targeted antimicrobial applications [50]. Comparisons with ascorbic acid as a benchmark for antioxidant performance revealed significant scavenging capabilities of Cu-GO in the DPPH assay. The nanocomposite’s antioxidative potential suggests its ability to neutralize free radicals, indicating a potential role in combating oxidative stress-related conditions and diseases. In the assessment of anti-obesity activity, Cu-GO demonstrated the inhibition of pancreatic lipase enzyme activity, showcasing its potential as an anti-obesity agent. A comparison with Orlistat, a known pharmaceutical agent for weight management, establishes the relative efficacy of Cu-GO, prompting discussions on mechanisms, safety, and biocompatibility. Further research avenues may explore the optimal conditions, potential synergies, and overall therapeutic potential of Cu-GO. The discussion on anti-obesity activity aims to provide a comprehensive understanding of experimental findings and the implications for Cu-GO’s development as a potential anti-obesity agent.

The collective evidence from antimicrobial, antioxidant, and anti-obesity assessments positions Cu-GO as a versatile nanocomposite with potential applications in healthcare and related industries. While promising, further research is warranted to elucidate the mechanisms, optimize the dosage, and assess the long-term safety, providing a comprehensive understanding of Cu-GO’s potential advantages and limitations in biomedical contexts. This study establishes Cu-GO as a promising nanocomposite with antimicrobial, antioxidant, and anti-obesity activities. The multifunctional properties of Cu-GO suggest diverse applications in healthcare and underscore the importance of continued exploration and development for biomedical purposes. Graphene, with its expansive surface area and distinctive optical, transport, mechanical, and electronic properties, emerges as a promising candidate for diverse applications, spanning solar cells, sensors, hydrogen generation, and catalytic activities. In the domain of wastewater management, reduced graphene oxide (RGO) and its composites, notably those involving visible light, have gained prominence for the degradation of both cationic and anionic dyes, as well as the removal of toxic Cr(IV) from aqueous solutions.

The synthesis of graphene- or graphene-oxide (GO)-based nanocomposite photocatalysts has become a growing trend, leveraging their unique properties for the effective degradation of organic dyes [31,32]. This research specifically investigates the effectiveness of the Cu-GO composite as a sunlight-active photocatalyst, focusing on the degradation of the RhB dye. Numerous studies underscore the potential of heterostructures and composite materials in enhancing the photodegradation of various dyes, including RhB. The uncontrolled release of RhB into water poses environmental and public health concerns, necessitating innovative approaches for wastewater treatment. By exploring the visible-light-induced photocatalytic capabilities of the Cu-GO composite, this study aims to contribute to the development of efficient methodologies to address the challenges associated with organic dye degradation, mitigating the adverse impact on the environment and public health. The observed high adsorption efficiency (53%) of the Cu-GO nanocomposite is attributed to its unique properties and structural characteristics. The combination of copper oxide and graphene oxide enhances the adsorption capacity due to their synergistic effects, providing specific adsorption sites and promoting interactions through π-π stacking and van der Waals forces [48].

In our recent study, we investigated the performance of metal oxide photocatalysts under sunlight irradiation [51]. Comparing the photodegradation effectiveness of Cu-GO with various metal oxides, we observed the following trend: Cu-GO (73.8%) > CuO (41.5%) > NiO (28.2%) > TiO_2_ (17.2%) > SnO_2_ (14.8%). This comparison highlights the superior performance of Cu-GO as a photocatalyst. Additionally, the efficiency of GO-based materials in absorbing visible light, attributed to their lower band gap energy of 1–2 eV, underscores their effectiveness as sunlight-active photocatalysts compared to commercial metal oxide counterparts.

The high adsorption efficiency of Cu-GO leads to the anticipation of correspondingly elevated photocatalytic efficiency. The efficient adsorption of RhB onto the Cu-GO composite surface suggests a promising potential for enhanced degradation under light irradiation, emphasizing the crucial role of material characteristics in influencing overall photocatalytic performance. Notably, the Cu-GO composite demonstrated remarkable efficiency at 73.8% for RhB degradation under sunlight irradiation, outperforming graphite and GO. These findings underscore the superior photocatalytic performance of the Cu-GO composite, highlighting its potential as an efficient material for RhB degradation. The significant enhancement in efficiency under sunlight irradiation emphasizes the versatility and robustness of Cu-GO as a promising photocatalytic material for sustainable environmental remediation processes.

## 5. Conclusions

In this investigation, we successfully synthesized the biologically active copper(II) complex [Cu(im)_2_]Cl_2_ using a well-established method. After its synthesis, we strategically integrated this copper(II) complex onto graphene oxide, resulting in the creation of a nanocomposite termed copper(II)-complex-grafted graphene oxide (Cu-GO). The extensive characterization of Cu-GO involved multiple techniques, including X-ray diffraction (XRD), Fourier-transform infrared spectroscopy (FT-IR), UV–visible spectroscopy, emission spectra analysis, X-ray photoelectron spectroscopy (XPS), and X-ray absorption fine structure (XAFS). The antibacterial potential of the Cu-GO compounds was systematically evaluated through disk diffusion and micro broth dilution methods. Significantly, the copper complex emerged as the most potent, displaying a Minimal Inhibitory Concentration (MIC) value against *Klebsiella* bacteria. The comparative screening of the antibacterial activities highlighted the superior performance of the copper-complex-grafted GO when compared to the standalone copper complex. Expanding our investigations, we explored the antioxidant and anti-obesity activities of the copper-complex-grafted GO against *Staphylococcus* organisms. The outcomes revealed promising prospects for further exploration into the diverse health-related applications of these compounds. Turning our attention to the photocatalytic domain, we assessed the Cu-GO nanocomposite under sunlight irradiation. The results not only showcased enhanced biological activity, but also indicated a noteworthy level of moderate photocatalytic performance. This dual functionality underscores the potential versatility of the Cu-GO nanocomposite, offering heightened biological efficacy coupled with controlled photocatalysis. In summary, our study provides valuable insights into the multifunctional properties of copper(II)-complex-grafted graphene oxide nanocomposites. The demonstrated effectiveness in antibacterial, antioxidant, and anti-obesity activities, coupled with moderate photocatalytic performance, opens up new avenues for their application across diverse fields. The versatile nature of Cu-GO suggests its potential significance in advancing innovative solutions and applications, marking it as a promising candidate for a range of biomedical and environmental applications.

## Figures and Tables

**Figure 1 materials-17-00892-f001:**
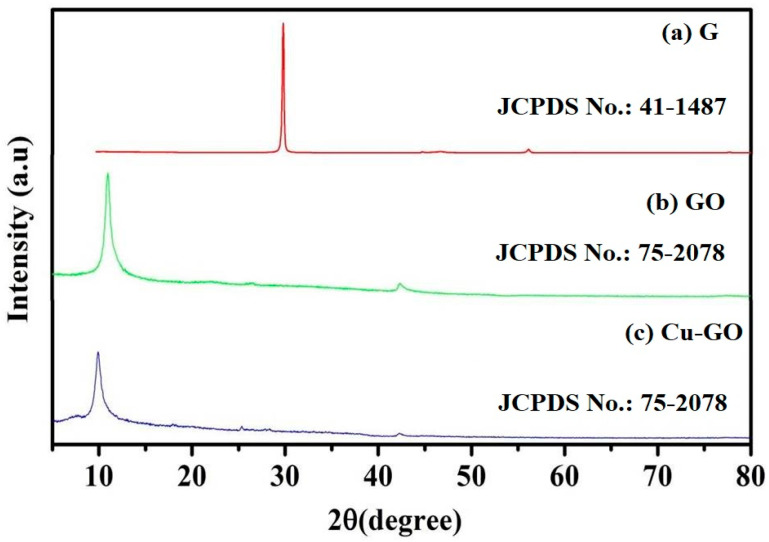
X-ray diffraction (XRD) pattern for (**a**) graphite (G), (**b**) graphene oxide (GO), and (**c**) the Cu-GO composite (note: “a.u.” is arbitrary units).

**Figure 2 materials-17-00892-f002:**
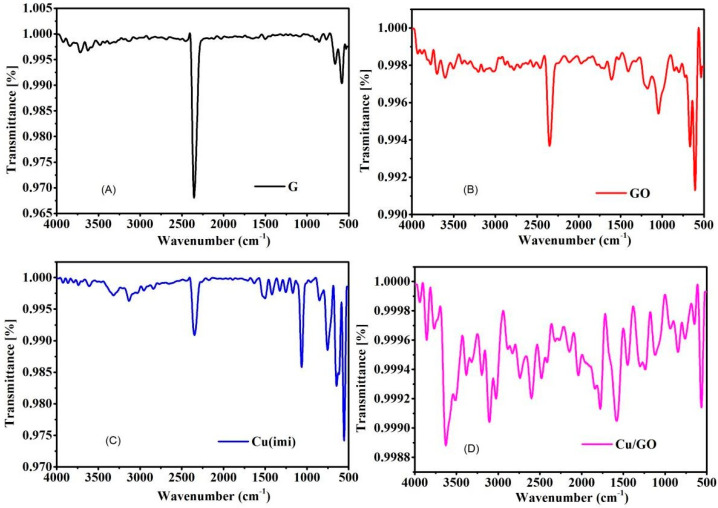
Fourier-transform infrared (FT-IR) spectra for (**A**) graphite (G), (**B**) graphene oxide (GO), (**C**) precursor copper(II)–imidazole complex, and (**D**) Cu-GO composite.

**Figure 3 materials-17-00892-f003:**
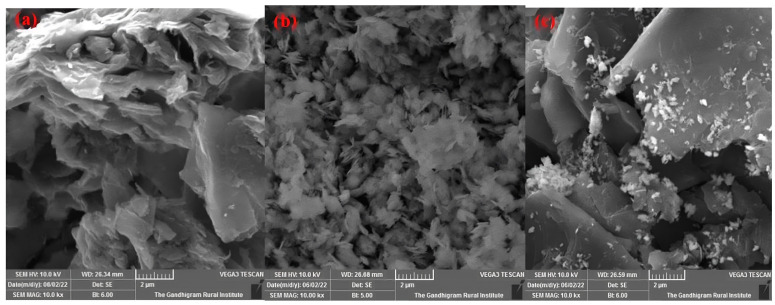
Scanning Electron Microscopy (SEM) images of (**a**) graphite, (**b**) graphene oxide, and (**c**) Cu-GO nanocomposite.

**Figure 4 materials-17-00892-f004:**
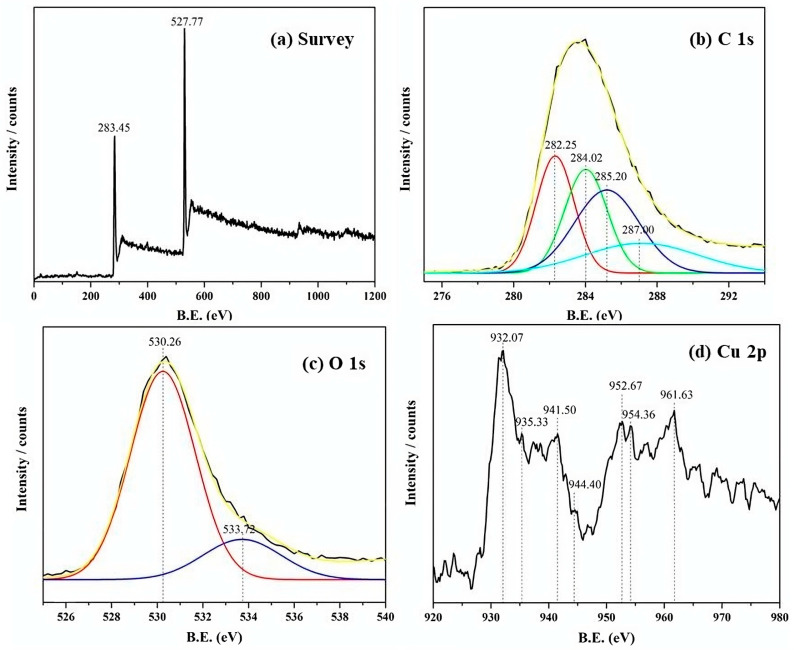
X-ray Photoelectron Spectroscopy (XPS) spectra illustrating the Cu-GO nanocomposite: (**a**) total survey spectra, (**b**) C 1s spectra (Gaussian fitting peaks shown in different colors), (**c**) oxygen spectra (Gaussian fitting peaks shown in different colors), (**d**) Cu 2p spectra. Note: The solid black line represents the data, while the other colored lines represent the fitted peaks.

**Figure 5 materials-17-00892-f005:**
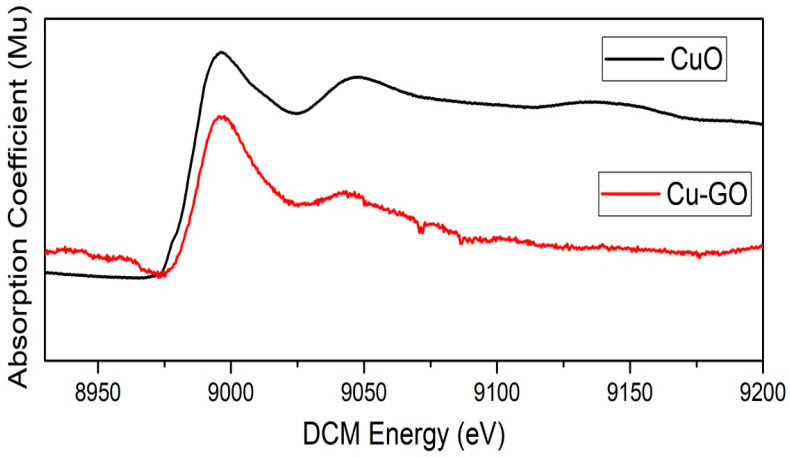
X-ray Absorption Near Edge Structure (XANES) spectra at the K-edge for copper, comparing CuO and Cu-GO nanocomposite.

**Figure 6 materials-17-00892-f006:**
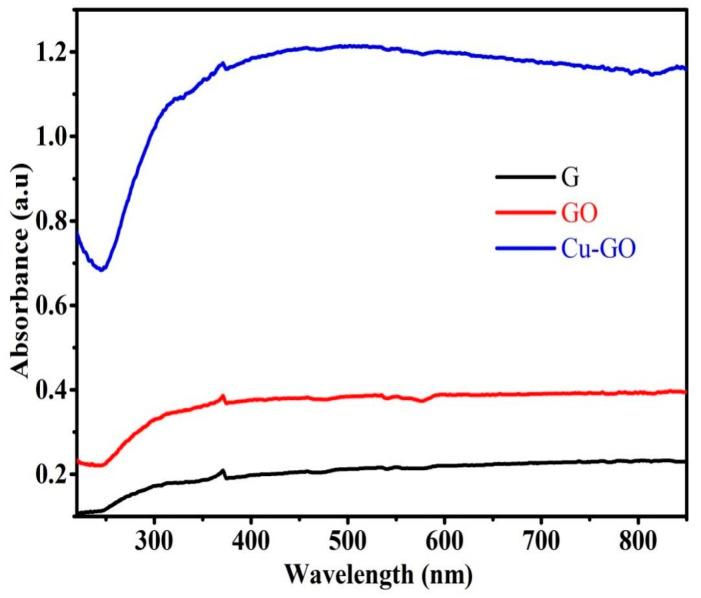
Ultraviolet diffuse reflectance spectra (UV-DRS) for graphite (G, black line), graphene oxide (GO, red line), and Cu-GO composite (Cu-GO, blue line).

**Figure 7 materials-17-00892-f007:**
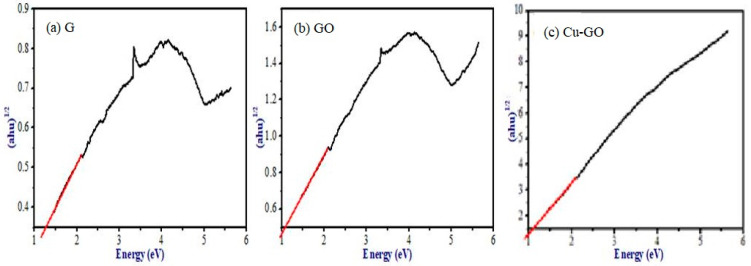
Tauc plot images of (**a**) graphite (G), (**b**) graphene oxide (GO), and (**c**) Cu-GO for the calculation of the band gap energy. Note: The solid black line represents the data, while the red lines represent the fitted line.

**Figure 8 materials-17-00892-f008:**
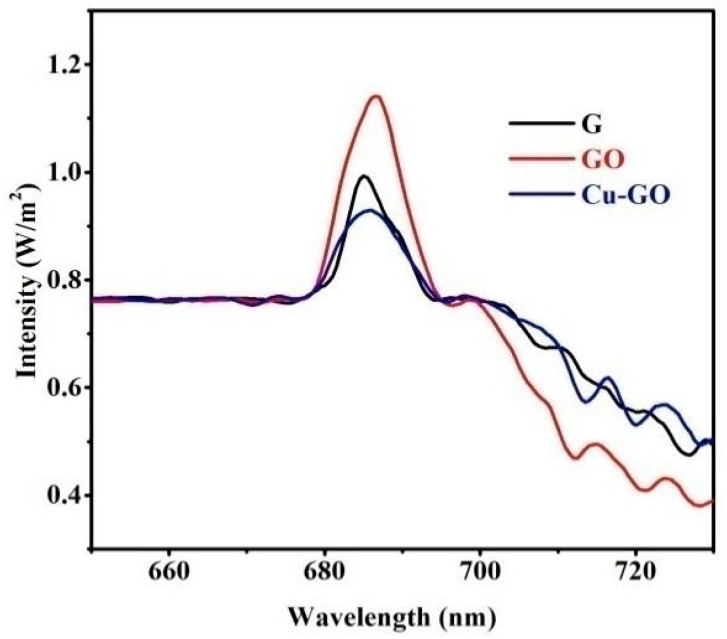
Steady-state emission spectra of graphite (G, black line), graphene oxide (GO, red line), and Cu-GO composite (Cu-GO, blue line) at λ_exc_ = 458 nm.

**Figure 9 materials-17-00892-f009:**
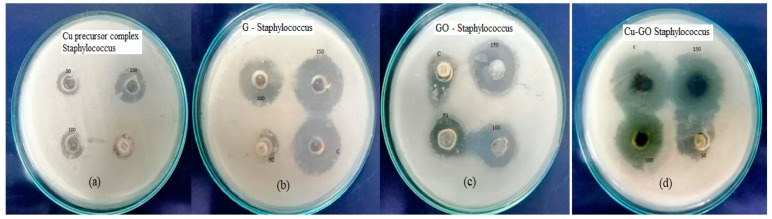
Images of organism: *Staphylococcus*, control: Ceftazidime antibiotic disks on (**a**) Cu-precursor complex, (**b**) graphite (G), (**c**) graphene oxide (GO), and (**d**) Cu-GO nanocomposite. Note: Varying concentrations of the samples (50 µL, 100 µL, 150 µL, and C the control).

**Figure 10 materials-17-00892-f010:**
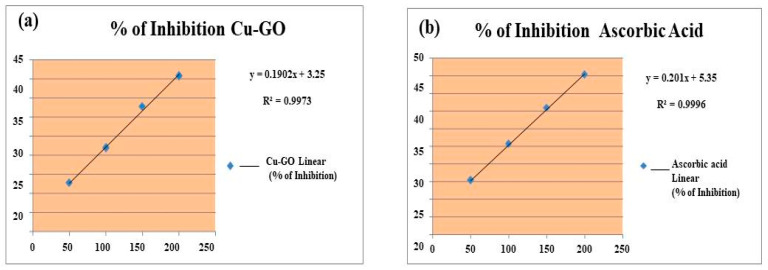
Illustrates the antioxidant activity, presented as the percentage of inhibition, for both (**a**) Cu-GO and (**b**) ascorbic acid, allowing for comparison between the two.

**Figure 11 materials-17-00892-f011:**
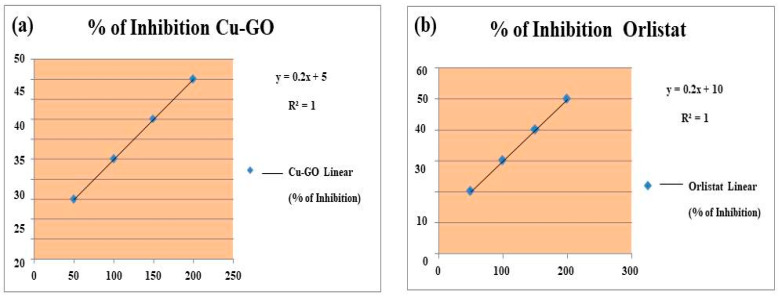
Anti-obesity activity, presented as the percentage of inhibition, for both (**a**) Cu-GO and (**b**) Orlistat, allowing for comparison between the two.

**Figure 12 materials-17-00892-f012:**
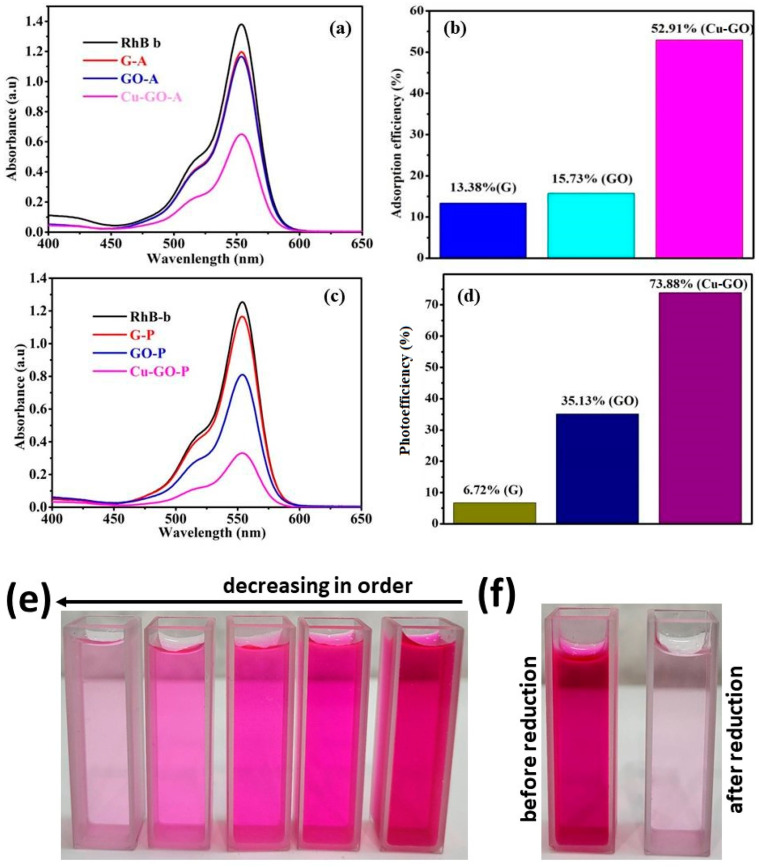
(**a**) UV–visible spectra of RhB adsorption solutions, (**b**) the corresponding adsorption efficiency, and (**c**,**d**) UV–visible spectra of RhB photo-irradiated solutions under sunlight with different catalysts; (**e**,**f**) digital photograph of images of decreasing concentrations of RhB solutions from right to left, (**f**) before and after reduction of RhB.

**Figure 13 materials-17-00892-f013:**
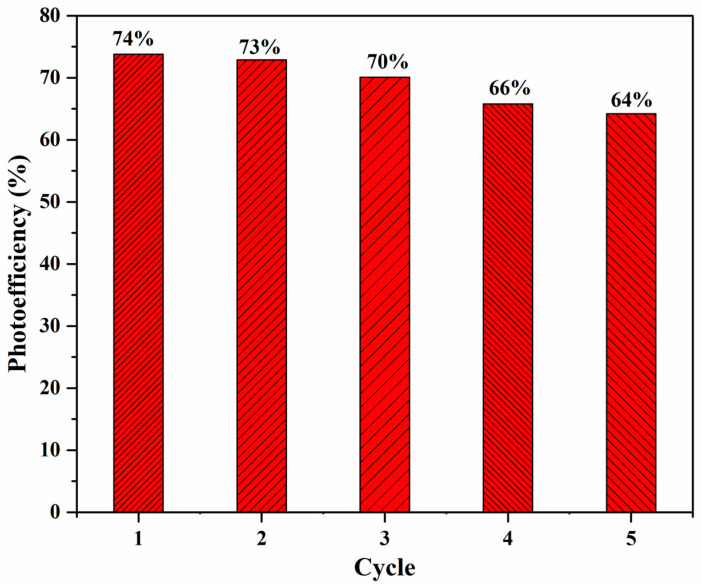
Photo-efficiency trends of the Cu-GO nanocomposite across successive catalytic cycles.

**Table 1 materials-17-00892-t001:** XRD data.

Sample ID	K	λ (Ȧ)	β	θ	D (nm)
G	0.94	1.5406	0.25019	13.2150	59.47
GO	0.94	1.5406	0.75321	6.4361	19.34
Cu-GO	0.94	1.5406	0.86792	4.9691	16.75

**Table 2 materials-17-00892-t002:** Zone inhibition of G, GO, Cu precursor complex, and Cu-GO composite (anti-microbial activity, organism: *Staphylococcus*, control: Ceftazidime antibiotic disk).

Sample ID	Zone of Inhibition (mm/mL)			
	50 µL	100 µL	150 µL	Control
Cu precursor complex	08	09	10	15
G	9	11	12	12
GO	10	12	14	15
Cu-GO	13	14	15	18

**Table 3 materials-17-00892-t003:** Antioxidant activity was assessed through DPPH assay, presenting the percentage of inhibition for Cu-GO.

Test	Concentration of the Sample (mg/mL)	% of Inhibition Cu-GO	Ascorbic Acid
DPPH assay	50	12.7	15.2
	100	21.9	25.6
	150	32.7	35.8
	200	40.8	45.3
IC50 value	-	246.05	222.13

**Table 4 materials-17-00892-t004:** Anti-obesity activity was measured through pancreatic lipase enzyme inhibition, indicating the percentage of inhibition for Cu-GO.

Test	Concentration of the Sample (mg/mL)	% of Inhibition Cu-GO	Orlistat
Pancreatic Lipase enzyme activity	50	15	20
	100	25	30
	150	35	40
	200	45	50
IC50 value	-	225	200

**Table 5 materials-17-00892-t005:** Comparison of catalytic activities for the reduction in RhB reported in the literature.

Catalytic Name	Rate Constant K(min^−1^)/% Reduction	Reference
Graphdiyne–Zinc Oxide (GD-ZnO)	0.00298	[33]
Nano-hybrid	0.02263	[34]
ZnO–10% reduced graphene oxide	NA	[35]
ZnO-reduced graphene oxide nanocomposites	NA	[36]
ZnO nanoparticle	NA	[37]
Graphene–zinc oxide nanocomposites	NA	[38]
ZnO NW/RGO nanocomposites	NA	[39]
ZnO nanorods	0.029	[40]
ZnO/AgBr/Fe_3_O_4_/Ag_3_VO_4_	0.0034	[41]
Graphene oxide	35.1%	This study
Cu-GO nano-composite	73.8%	This study

Where NA: Not available.

## Data Availability

The data that support the findings of this study are available from the corresponding author upon reasonable request.

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
