# Peer review of "Fine-Tuned Graphene Oxide Nanocomposite: Harnessing Copper(II)–Imidazole Complex for Enhanced Biological Responses and Balanced Photocatalytic Functionality"

_materials, 2024, doi:10.3390/ma17040892_

Round 1

Reviewer 1 Report

Comments and Suggestions for Authors

When presenting their results, the authors often speak in general terms. It would be much better if they emphasized the specifics and explained their experimental results in more detail. The value of the article can be improved by significantly revising the text

1.       „Crystallography studies were conducted on the prepared nickel(II) complexes using a rotating anode diffractometer 116 (XRD-PHILIPS-PANALYTICAL, Netherlands) with Cu-kα irradiation”- This sentence needs to be corrected: prepared copper(II)complexes.

2.       How was the data shown in table 1 calculated? I consider it necessary to indicate the meaning of the various parameters!

3.       A more detailed analysis of the FTIR spectra is required. If possible, the more characteristic peaks should be explained. It should be indicated what conclusions can be drawn from their changes!

4.       The 4.b. and 4.c. the figures do not indicate the meaning of the spectrum lines of different colors.

5.       „For copper(II), one would anticipate Cu2p3/2 and Cu2p1/2 peaks, as depicted in Figure 4d”- The identification of the peaks should be indicated.

6.       „Sample G exhibits broad absorption peaks around 372 and 458 nm. In contrast,  sample GO displays sharp peaks at 272 and 296 nm, accompanied by a broader peak spanning 453-462 nm. Sample Cu-GO reveals two distinct broad peaks around 343-361 nm and 420-430 nm, along with a smaller, sharper peak at 592 nm.” - In which figure can you see these bands? Unfortunately, this is not clear to me.

7.       In Figure 8 (left), it is difficult to distinguish the data for each material. Perhaps the error can be eliminated by changing the assignment of the y axis (e.g. 0-8).

8.       Line 648: ” The decrease in band gap energy for GO and Cu-GO compared to G powder indicates the efficacy of our preparation method in creating a visible-light active photocatalyst. Consequently, our photocatalytic studies were conducted under sunlight irradiation.”- Sunlight contains only a few % of UV light, but its effect on the photocatalytic effect is not negligible. Therefore, it is not correct to consider sunlight and visible light as the same. The sunlight used in the experiments was not characterized. The TiO2 catalyst also works when illuminated by sunlight.

9.       line 661 „Moreover, the identification of emission peaks and their intensities contributes to a deeper understanding of the photoinduced phenomena in the studied samples.”- No peaks were identified here. Only general statements can be read.

10.    line 721 „This research specifically investigates the efficacy of the Cu-GO composite as a visible-light-induced photocatalyst, targeting the degradation of the cationic RhB dye”- The statement is incorrect.

Comments on the Quality of English Language

The language of the article is correct.

Author Response

The responses to the reviewer's comments are attached in a separate Word document

Reviewer 2 Report

Comments and Suggestions for Authors

In this work, the authors synthesized copper(II) complex grafted graphene oxide and comprehensively characterized the structure by using several techniques. Before being accepted for publication, the submitted manuscript needs a major revision based on the following comments:

1.      Could you provide a more detailed emphasis on the novelty of the manuscript to better communicate its unique contributions?

2.      It is recommended to explicitly articulate the significance of the paper within the Introduction section. Could you elaborate on the key points that highlight its importance?

3.      The use of "a.u." as an abbreviation for arb. units is not suitable, as it is commonly associated with astronomical units. It is advisable to select an alternative abbreviation. Could you consider making this adjustment?

4.      Given the wide array of bacteria globally, why did the author specifically choose Klebsiella bacteria? Could you provide a rationale or explanation for this selection?

5.      To enhance reader engagement, it is suggested that the abstract section include some qualitative good results. Could you elaborate on the specific results that would contribute to a more impactful abstract?

6.      In Figure 1, the inclusion of hkl values and corresponding JCPDS file numbers in the XRD pattern image is appreciated. However, it would be beneficial if the author could explain the shift in peaks after the addition of Cu. Additionally, please incorporate this explanation into the revised manuscript's discussion.

7.      The discussion of the FTIR study is deemed insufficient. Could the author provide a detailed explanation for each peak observed in the FTIR spectra?

8.      The energy bandgap value calculated in the tauc plot image needs reconsideration, especially if there are no observed changes in bandgap values. Could the author provide additional insights into the calculation and its consistency?

9.      Regarding the stability of the photocatalyst through recycling, it is suggested that the information be reiterated at least three times. Additionally, after each recycling, could the author analyze the structure and functional groups of the photocatalyst for consistency?

10.  To substantiate the claim of RhB degradation, it is recommended to conduct COD and BOD5 experiments. Could the author consider including these experiments to verify the asserted degradation of RhB?

Comments on the Quality of English Language

1.      Typographical errors are present throughout the manuscript. Authors are required to pay keen attention to this.

Author Response

(The authors gave the same response as above.)

Round 2

Reviewer 1 Report

Comments and Suggestions for Authors

I accept the authors' answers and corrections.

Reviewer 2 Report

Comments and Suggestions for Authors

In revised form, this manuscript justified all comments and has improved significantly to justify a publication